# Spatial Analysis of the Ecogeographic Diversity of Wild Creeping Cucumber (*Melothria pendula* L.) for In Situ and Ex Situ Conservation in Mexico

**DOI:** 10.3390/plants13182572

**Published:** 2024-09-13

**Authors:** Rosalinda González-Santos, Luis Hernández-Sandoval, Mauricio Parra-Quijano

**Affiliations:** 1Laboratorio de Botánica, LANIVEG, Facultad de Ciencias Naturales, Universidad Autónoma de Querétaro, Avenida de las Ciencias S/N, Juriquilla, Querétaro C.P. 76230, Mexico; rosalinda.gonzalez@uaq.mx (R.G.-S.); luishs@uaq.mx (L.H.-S.); 2Facultad de Ciencias Agrarias, Universidad Nacional de Colombia, Sede Bogotá, Ciudad Universitaria, Bogotá 14490, Colombia

**Keywords:** representativeness, underutilized species, hotspot, ELC maps, abiotic adaptation

## Abstract

*Melothria pendula* L., a wild relative of cucurbit crops, is also used for food and as a medicinal plant in Mexico. The objective of this study was to ecogeographically characterize the known populations of *M. pendula* in Mexico, determining its adaptive range and possible sites for in situ and ex situ conservation. To achieve this goal, we compiled a dataset of 1270 occurrences of *M. pendula* from herbarium and botanical databases and individual observations. Adaptive scenarios were generated through the development of an ecogeographic land characterization (ELC) map, preceded by the identification of abiotic variables influencing the species’ distribution. Eleven bioclimatic, edaphic, and geophysical variables were found to be important for the species’ distribution. The ELC map obtained contained 21 ecogeographic categories, with 14 exhibiting the presence of *M. pendula.* By analyzing ecogeographic representativeness, 111 sites of high interest were selected for the efficient collection of *M. pendula* in Mexico. Eight high-priority hotspots for future in situ conservation of *M. pendula* were also identified based on their high ecogeographic diversity, with only three of these hotspots located within protected natural areas. In this study, ecogeographic approaches show their potential utility in conservation prioritization when genetic data are scarce, a very common condition in crop wild relatives.

## 1. Introduction

*Melothria pendula* L., known as “sandiita” in Spanish, is a member of the Cucurbitaceae that has been identified as an interesting option to strengthen local and regional food and nutritional security in Mexico [1,2,3]. Its nutritional value stands out due to its high content of protein (12.6%), fiber (16.3%), and carbohydrates (56.8%) [4]. While mainly utilized for human consumption in Mexico, some indigenous communities attribute medicinal properties to *M. pendula*, and it is also used as fodder for animals [5]. Although this species is not cultivated for commercial purposes, farmers let it grow in or around homes, harvesting its fruits from wild, weedy, and ruderal populations [5]. Given its wide range of uses and presence in different ecological strata, *M. pendula* is considered a wild species in an early stage of domestication [6].

The species is native to America, and its distribution spans from the United States to northern Argentina [7]. It can be found from sea level to over 2000 m.a.s.l., in areas with humid and sub-humid climates; however, it has also been found in arid and semi-arid climates [3,6,7]. *Melothria pendula* is monoecious and can reproduce via seed and vegetative propagation. Its plants are perennial and herbaceous with a creeping and climbing growth habit [1,4]. This species is a wild relative of the genera *Cucurbita*, *Sechium*, *Cucumis*, and *Citrullus*, all Cucurbitaceae, but sharing the same tribe (Benincaseae) only with *Cucumis* and *Citrullus* [8].

When addressing the conservation of biodiversity of wild plants of agricultural interest, especially those with limited genetic studies and genotyping data across many populations, an ecogeographic approach has proven effective in ensuring representativeness in their ex situ conservation. Ecogeography has emerged as a useful alternative to expand knowledge on the adaptive potential of plant genetic resources and to identify functional genetic traits of high interest [9,10]. A straightforward application of ecogeography is the creation of ecogeographic land characterization (ELC) maps. These maps are used to identify different adaptive scenarios that a species may occupy within its spatial distribution by collecting data for known occurrence sites. The development of ELC maps considers bioclimatic, edaphic, and geophysical variables, which are key for plant adaptation and development [11,12]. By overlaying the spatial distribution of points representing the localities where a species has been observed on an ELC map, it is possible to determine the species’ adaptive range and the ecogeographic representativeness of a set of ex situ conserved accessions [13,14,15].

It is also possible to conduct ecogeographic characterization of each population of the species, rather than of the territory, based on a series of variables that represent aspects of its abiotic adaptation extracted from its location. This last methodology can be considered a direct characterization of a set of populations, while an ELC map constitutes an indirect characterization through a prior characterization of the territory where the species is distributed. Both approaches are closely related to abiotic adaptation processes, which are, in turn, reflected in functional genetic diversity [16,17]. This knowledge has proven useful in the optimized design of germplasm surveys and collections, facilitating the creation or improvement of genebanks of both cultivated and wild plant species. The optimized collecting design (OCD), introduced by Parra-Quijano et al. [18], proposes to create or enhance genebanks through germplasm collections guided by the concept of ecogeographic representativeness and its association with adaptive genetic representativeness. For example, Marinoni et al. [19] applied this approach to a native wild species with forage use, *Trichloris crinita*, to improve the representativeness of its germplasm collection.

Methodologies have been developed to enhance the efficiency of plant germplasm collection for ex situ conservation purposes and to determine priority areas for in situ conservation of plant genetic resources for food and agriculture, even without prior knowledge about the genetic status of the target species’ populations. To identify genetic diversity hotspots within the distribution of species characterized by molecular markers, van Zonneveld et al. [20] developed a spatial analysis technique based on the determination of genetic diversity indicators, such as heterozygosity, calculated by geographic neighborhoods. These genetic diversity hotspots constitute ideal areas for in situ conservation of plant genetic resources since it is presumed that greater diversity can be conserved in a smaller area [21,22]. Building on this idea, Parra-Quijano et al. [23] proposed using mean Euclidean distances between georeferenced populations characterized ecogeographically by geographic neighborhoods. This approach generates a map of ecogeographic diversity that helps identify hotspots for such diversity. A geographic analysis of genetic diversity, or its ecogeographic surrogate, can assist in making decisions related to in situ conservation [24].

Despite the considerable potential of *M. pendula* for food and agriculture in Mexico and other countries, there is a lack of information regarding (i) representative data on genetic diversity across its geographic distribution, (ii) its adaptive capacity and the location of populations with potential adaptations of interest for plant breeding, and (iii) sites of special interest for ex situ and in situ conservation. Additionally, no genebanks have been reported for this species. Providing this knowledge for a species that currently has no significant genebanks beyond 12 accessions registered in the Genesys portal [25] and that lacks in situ conservation plans in any country can be key to supporting the decision-making processes of authorities regarding its future conservation. Therefore, this study aimed to ecogeographically characterize known and georeferenced populations of *M. pendula* in Mexico to develop an optimized collecting design. This can facilitate the establishment of the first genebank of high representativeness for this species and potential in situ conservation sites of high adaptive diversity.

## 2. Results

### 2.1. Distribution of M. pendula

The quality analysis performed by the GEOQUAL tool shows that of the 1270 initial records, 1063 (83.7%) exceeded the defined quality threshold. A greater density of occurrences was observed near the Gulf of Mexico and on the Yucatan Peninsula, with lower densities in the west and north of Mexico. The 50 × 50 km cells with the highest number of records are in Veracruz, the Yucatan Peninsula, and Chiapas, located towards the Gulf of Mexico, with a value of 64 (Figure 1).

### 2.2. Ecogeographic Variables of Importance

Of the 179 available ecogeographic variables, 48 were retained in the subjective variable selection process, distributed as follows: 12 bioclimatic variables depicting maximum and minimum temperature and precipitation; 19 edaphic variables related to soil texture, nutrient availability, and pH; and 17 geophysical variables including elevation, slope, northness, eastness, and annual solar radiation (Table A1). Eleven variables were finally selected in the objective variable selection process, with a variable retention rate of 6.14% for both selection processes. Table 1 shows the distribution of the variables that were finally selected across the ecogeographic components.

### 2.3. Ecogeographic Adaptive Scenarios

The ecogeographic land characterization, performed with the ELCmapas tool using the 11 variables selected in the previous step, produced an ELC map with 21 categories (adaptive scenarios) for Mexico (Figure 2). These 21 categories result from combining 3 bioclimatic, 4 geophysical, and 4 edaphic groups. Of the 48 potential categories that could emerge from this combination of groups, 27 are not reflected on the map since the combination that produces them does not occur in Mexico. The 1063 records of *M. pendula* are distributed across 14 of the 21 ELC map categories, with no presence in categories 2, 5, 12, 15, 18, 19, or 21.

### 2.4. A Design for a Representative Germaplsm Collection 

The Representa tool classified the ELC map categories by quartiles of frequency of occurrence as follows: low frequency with four categories (3, 6, 9, and 13), medium-low frequency with three categories (7, 10, and 11), medium-high frequency with three categories (14, 17, and 20), and high frequency with four categories (1, 4, 8, and 16) (Table 2). Additionally, 7 categories (2, 5, 12, 15, 18, 19, and 21) out of the 21 included in the null frequency group remain outside this classification. The high-frequency categories are located towards the coast of the Gulf of Mexico, southern Mexico, and the Yucatan Peninsula. Notably, the null frequency categories and those in the low-frequency group occur mainly in north-central Mexico (Figure 3).

By applying the proportional (P) strategy and a sampling intensity corresponding to 10% of the total known georeferenced populations, it was determined that the initial size of the germplasm collection of *M. pendula* should be 106 populations. Therefore, sampling should aim to reach this figure. As expected, the distribution of the total number of populations with georeferenced location information, compiled by frequency quartile groups, shows higher percentages in the high (863 populations for 86%) and medium-high (170 populations for 16.9%) frequency groups. Meanwhile, 26 populations (2.6%), and four populations (0.4%) were assigned to the medium-low and low-frequency groups, respectively. The exact numbers of populations to be collected by category, or quotas, are detailed in Table 2. To ensure that all categories were represented, we needed to increase the total number of collections to 111. With this adjustment, the representation (in percentage) of the medium-low and low-frequency groups increased by 1.2 and 3.2%, respectively, while the representation of medium-high and high categories decreased by 0.7 and 3.7%, respectively.

After randomly selecting populations to visit, based on the final quotas for each category, they are mainly located near the Gulf of Mexico and the southern region of the country, with fewer in the western areas (as mapped in Figure 4).

### 2.5. Map of Ecogeographic Diversity

The map of mean ecogeographic distances by geographical neighborhoods is shown in Figure 5. The hotspots of high ecogeographic diversity are distributed as follows: zone 1, Sierra Madre Oriental (SMO); zone 2, between the edge of the SMO and Oaxaca; and zone 5, in mountainous areas of Chiapas. Zones 3, 4, 6, 7, and 8 of medium-high diversity are located on the Yucatan Peninsula and near the Pacific Ocean. The highest values of mean ecogeographic distance per cell were 58, 61, and 54 for zones 1, 2, and 5, respectively. Of the hotspots of high diversity, zones 1 and 5 partially coincide with protected areas on the official SEMARNAT-CONANP map [26]. Zone 1 partially overlaps with the Sierra Gorda protected area, whereas zone 5 minimally overlaps with the protected areas of Montes Azules (Selva Lacandona) and Lacan-Tun. Zones 3 and 7 of medium-high diversity are completely included in the protected areas of Tuxtlas and Chamela-Cuixmala. However, zones 2, 4, 6, and 8 do not coincide with protected natural areas.

## 3. Discussion

*Melothria pendula* L. is an alternative food source for the rural communities where it is found [3,4,6,7]. Using herbarium records, 111 collecting sites were identified to establish a genebank with high representativeness of the different ecogeographic scenarios to which the species has adapted (Figure 4). This process involves the use of an optimized germplasm collecting design (OCD), a methodology that has been developed since 2005 with the creation of the first ELC map and the subsequent description of the ecogeographic representativeness concept [18]. Over time, elements such as predictive species distribution models [14], complementarity analysis for multispecies collection [27], and predictive characterizations [15] have been integrated into the concept of ecogeographic representativeness within OCD.

Usually, OCDs have been applied to improve existing genebanks; in this study, an OCD is being used to establish a new genebank for *M. pendula*. Besides contributing to the ex situ conservation of the species, it was possible to identify ecogeographic diversity hotspots using essentially the same initial information, contributing to in situ conservation. The approach used in this study is based on the spatial genetic characterization conducted in *Annona cherimola* by van Zonneveld et al. [20]. However, species of lesser economic importance, such as *M. pendula*, lack genetic characterization data. This underscores the importance of the ecogeographic approach proposed in this study, which was validated first by Hanson et al. [16] and later by Di Santo and Hamilton [17].

The approaches employed here for the effective conservation of the genetic diversity of *M. pendula* use a type of characterization that is neither genotypic nor phenotypic but rather focuses on adaptive characteristics, particularly abiotic ones, based on ecogeographic characterization, which can be of two types: characterizing each collecting site or occurrence, or characterizing the entire spatial framework [28]. In this study, both techniques were used in a complementary manner, the first for in situ conservation and the second for ex situ conservation. Several studies indicate the importance of both types of ecogeographic characterization to know the different scenarios in which the species has adapted, specifically identifying those sites associated with abiotic-stress conditions that may be relevant for future crop improvement [29]. Such analyses should be conducted before initiating in situ and ex situ conservation activities to optimize time and costs, and ensure the capture of ecogeographic representativeness and, indirectly, functional genetic diversity [30].

Regarding the OCD achieved through the ecogeographic characterization of the Mexican territory, this approach can identify relevant ecogeographic categories that are not represented in genebanks [19,31,32,33]. Additionally, the use of numerous bioclimatic, edaphic, and geophysical variables enables the definition of specific ecogeographic variables and categories for each species. For example, in the case of *M. pendula*, 11 variables determined the distribution, and 14 ecogeographic categories were defined. In a similar study on *Carica papaya*, with a distribution resembling that of *M. pendula*, 15 variables determined the distribution and 16 ecogeographic categories were defined [33]. Both studies only used herbarium records. However, Tapia et al. [32] demonstrated in an analysis of maize in the Andean region of Ecuador that the areas defined as priorities for in situ conservation, using only record information, matched areas identified through morphological characterization of germplasm accessions.

OCD designs are intended to build the most representative genebanks possible based on existent or available occurrence data and derived ecogeographical analyses. However, biases are well known in both species’ distribution data and genebank sampling. Meyer et al. [34] described different types of spatial biases in plant occurrence data stored in global databases, while Hijmans et al. [35] documented the same phenomena in genebanks. Since OCD designs use occurrence data from biodiversity and genebank databases, their collecting site recommendations are likely biased despite any randomization protocols that may be introduced, as in the present study. However, in terms of ecogeographical representativeness, OCD designs tend to reduce biases and include all adaptive scenarios. Depending on the sampling strategy, some of the adaptive scenarios can be more represented than others in the genebank.

In situ and ex situ conservation are fundamental and complementary strategies for the conservation and sustainable use of biodiversity, as established by the Convention on Biological Diversity [36] and the second Global Plan of Action for Plant Genetic Resources for Food and Agriculture [37]. Therefore, the methodology developed in this study can be applied to other plant taxa with limited genotypic and phenotypic characterization data to develop conservation strategies.

Regarding in situ conservation, the areas of greatest ecogeographic diversity of plant genetic resources for food and agriculture do not generally coincide with protected natural areas (PNAs) since these are established for disparate purposes. In this case, of the eight priority areas, only three are located within PNAs (Figure 5). These results mirror those of Nduche et al. [38], who found that only 18.5% of occurrences of wild crop relatives in East Africa are in PNAs; a contrasting study of wild relatives in Malawi found that 66.8% are located within PNAs [39]. Sites that do not coincide with PNAs can be proposed as priority complementary sites, or recommendations made to expand PNA areas for in situ conservation.

## 4. Materials and Methods

### 4.1. Occurrence and Distribution Data

A database was compiled with 1000 records from the Global Biodiversity Information Facility [40], 253 records from the National Herbarium of Mexico (MEXU) of the Universidad Nacional Autónoma de México [41], and 17 records from the Querétaro Herbarium “Dr. Jerzy Rzedowki” (QMEX) [42]. To ensure robust results, the georeferencing quality of the collected occurrence data was evaluated with the GEOQUAL tool of CAPFITOGEN3 [23]. In their study, Khaki Mponya et al. [39] used a GEOQUAL threshold of 50 on a scale of 0 to 100. However, in the present study, we applied a threshold of 70 to identify occurrences with high uncertainty in their georeferencing (values from 0 to 69), which were discarded in subsequent analyses. An abundance map of records exceeding the quality threshold was then produced on a grid of 50 × 50 km cells.

### 4.2. Selection of Ecogeographic Variables of Importance

The selection of ecogeographic variables with major influences on the distribution of *M. pendula* was conducted through an initial subjective step followed by an objective step. In the subjective step, initial filtering was performed on a list of 179 ecogeographic variables, available for download from the CAPFITOGEN3 website [43]. These variables were provided in .tif format (GIS layer) and adjusted for resolution, coverage, and coordinate system to work with the CAPFITOGEN3 toolkit, which was cropped for many countries, including Mexico. From this list, irrelevant variables were eliminated based on a literature review of the species’ adaptation and expert surveys.

The second filtering step (objective) was performed with the SelecVar tool of CAPFITOGEN3. This tool uses a product of random forest classification analysis [44] called mean decrease in accuracy (MDA) to estimate the importance of each variable. From the set of important variables, redundant ones were then eliminated through a bivariate correlation analysis in a stepwise process. A variable was considered correlated with another when the Pearson correlation value was >0.5 or <−0.5, with a *p* < 0.001. SelecVar was configured to discard 33% of the variables based on MDA (parameter percenRF = 0.66), and then 66% of those filtered by MDA through bivariate correlations (parameter percenCorr = 0.33). Additionally, the tool was set to select a minimum of three ecogeographic variables (parameter nminvar = 3) for each component (bioclimatic, geophysical, and edaphic) in the final process. The complete parameter settings specifically used for SelecVar are shown in Table A2.

### 4.3. Ecogeographic Land Characterization

Variables from SelecVar were used to obtain an ecogeographic land characterization (ELC) map specific to *M. pendula*. The ELCmapas tool of CAPFITOGEN3 was utilized following a set of parameters to produce a final map with a cell size of 2.5 arcminutes (approximately 5 km at the equator). The optimal clustering determination method of Calinski and Harabasz [45] (parameter metodo = “calinski”) was used, with a maximum number of clusters per ecogeographic component of eight (parameter maxg = 8), based on the environmental and physiographic heterogeneity conditions where *M. pendula* is distributed. ELCmapas combines the bioclimatic, geophysical, and edaphic groupings to generate a map that represents the different adaptive scenarios for *M. pendula* in Mexico in the form of categories. It also generates a table with a detailed description of each category based on central tendency statistics for each variable involved. The complete list of parameters used to set up the ELCmapas tool is shown in Table A3.

### 4.4. Optimized Germplasm Collecting Design

The CAPFITOGEN3 tool [23] was used to determine the distribution of *M. pendula* across the ELC map categories. This tool assigns an ELC map category to each occurrence based on its location, thus generating the distribution and frequency of *M. pendula* over the ecogeographic categories available in the territory. It then performs a Chi-square test comparing this distribution with the total frequency of the ELC map categories. A significant difference (*p* < 0.001) between both distributions indicates that *M. pendula* prefers certain environments and avoids others, with the opposite corresponding to a random distribution or no preferences. Additionally, Representa creates five groups by frequency of occurrence of the species across the ELC map categories. The groups correspond to a quartile partitioning of categories according to their frequency in the entire territory as follows: null frequency (which groups categories with no presence of the species), low frequency (first quartile), medium-low frequency (second quartiles), medium-high frequency (third quartile), and high frequency (fourth quartile) [18,23]. These groups identify the categories preferred or avoided by the species, which can be useful when assessing the adaptive representativeness of ex situ conservation. For example, populations occurring in the first group (low frequency), which are often absent from germplasm collections, are frequently a priority for representing extreme environments in the adaptation of the target species within its known distribution [14]. Finally, based on the distribution of known *M. pendula* occurrences across the ecogeographic categories, a proportional (P) allocation strategy [46] with a sampling intensity of 10% of the total known georeferenced populations was applied to define the number of populations (quota) for each category where the species occurs that should be represented in the planned germplasm collection. When calculating quotas for categories with a low frequency of occurrence, caution should be exercised since at least one accession should be maintained as a quota, even if this requires a larger sample size than initially specified. In this way, we can establish a germplasm collection with ecogeographic representativeness adjusted to the environmental preference of the species, while avoiding the exclusion of any category where the species is known to occur. Once the quotas for each category are defined, the populations to be visited per category are ideally selected in a random manner. However, it is reasonable to consider making some adjustments to this selection according to the ease of access and available resources for collecting germplasm. The complete list of parameters used in the Representa tool is shown in Table A4.

To reduce the risk of not locating target populations in the field due to factors such as urbanization, climate change, or land use changes, the first step in our design was to assess the georeferencing quality of the gathered occurrence sites. We assigned a quality score to occurrences that, according to global land use maps such as GlobCover (1 km resolution), are located in areas compatible with the presence of the intended plant type (cultivated or wild) for collection (see the GEOQUAL tool [23]). By filtering occurrences, only those that meet predetermined quality criteria can proceed to subsequent steps and be considered potential collection sites. While this design increases the likelihood of successfully locating the desired populations, field validation remains, to date, the only definitive way to confirm the occurrence of the target population.

### 4.5. Map of Ecogeographic Diversity and Hotspot Coverage

To identify hotspots of high ecogeographic diversity, we determined the squared Euclidean ecogeographic distance between populations of *M. pendula* by spatial neighborhoods. This spatial analysis requires two inputs: a direct ecogeographic characterization of the occurrences and their accurate georeferencing. With these inputs, it is possible to obtain an ecogeographic diversity map using the DIVmapas tool [23]. DIVmapas first performs a direct ecogeographic characterization of the georeferenced populations for a list of abiotic variables specified by the user. In this study, the variables used to characterize the populations were the same as those used to create the ELC map, due to their relevance to the species’ adaptation. For analysis, DIVmapas requires the definition of map cell size, which was set to 5 km for this study. This cell size was used to create a grid for Mexico. From the centroids of those cells with at least one record of a population of *M. pendula*, DIVmapas established a circular area of influence (buffer area) with a radius of 10 km, and all the populations located within this area formed a spatial neighborhood. DIVmapas obtains the mean squared Euclidean distance from the ecogeographic characterization data calculated from all the populations in each neighborhood. The tool then assigns the mean value of the geographic neighborhood to the cell from whose centroid the buffer area was plotted. In this way, the ecogeographic diversity map is constructed, highlighting hotspots, which help define priorities for the in situ conservation of this species. Hotspots were identified based on mean Euclidean distance values greater than 5.61 (medium-high) or 6.58 (highest diversity). These thresholds were set by using the quantile function in R [47] applied to the squared Euclidean distance values of the grid cells, with probability values of 0.90 and 0.95, respectively. The complete list of parameters used in DIVmapas is provided in Table A5. Finally, the ecogeographic diversity hotspots that are partially or entirely covered by protected natural areas (PNA) were identified. This was achieved by overlaying the vector map of PNAs of the Mexican Republic at a scale of 1:250,000 [26] with the previously identified ecogeographic hotspots.

## 5. Conclusions

Conducting a spatial analysis of the ecogeographic diversity of *M. pendula* allowed the identification of 111 target collecting points to establish a genebank with genetic representativeness for ex situ conservation. Additionally, eight priority areas for in situ conservation of this species were established. These results confirm the need to define specific areas for the conservation of plant genetic resources for food and agriculture. A priori spatial analyses for conservation prioritization reduce time inputs, optimize financial resources, and help ensure the capture of representative genetic diversity of the species under study. This model for the identification of conservation priorities, both in situ and ex situ, can be replicated in other species with limited genotypic and morphological studies but with significant potential for food and agriculture.

## Figures and Tables

**Figure 1 plants-13-02572-f001:**
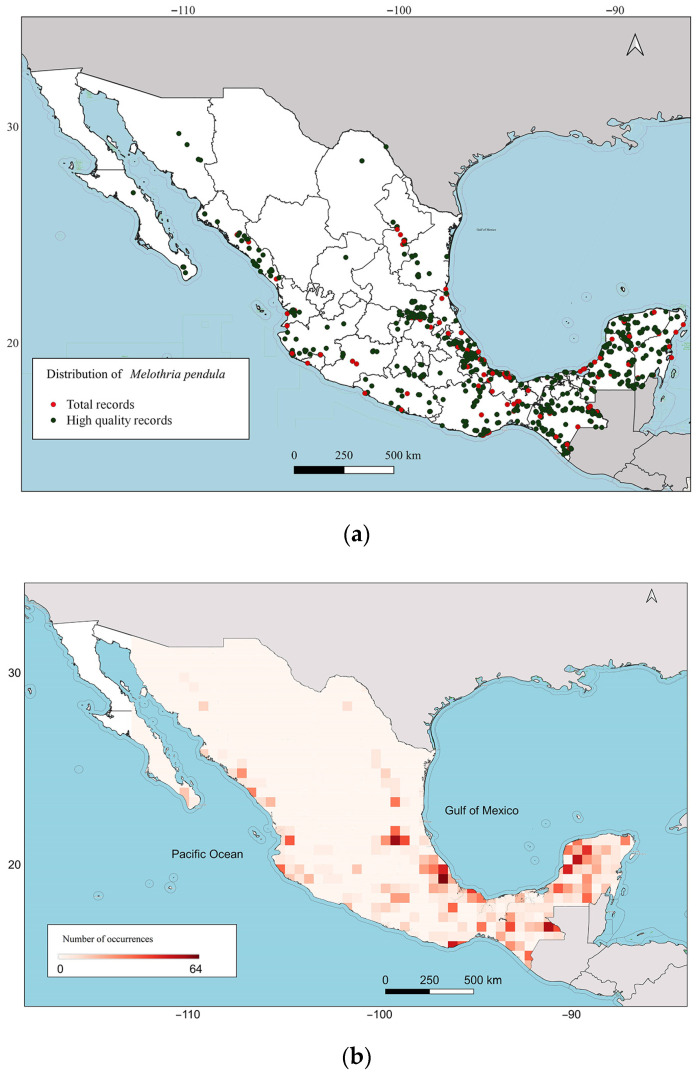
Distribution and abundance maps of high-quality occurrence data for *Melothria pendula* in Mexico. (**a**) Distribution of 1063 high-quality records identified by GEOQUAL (dark green dots) over 1270 total records (red dots). (**b**) Map of 50 × 50 km cells showing abundance values.

**Figure 2 plants-13-02572-f002:**
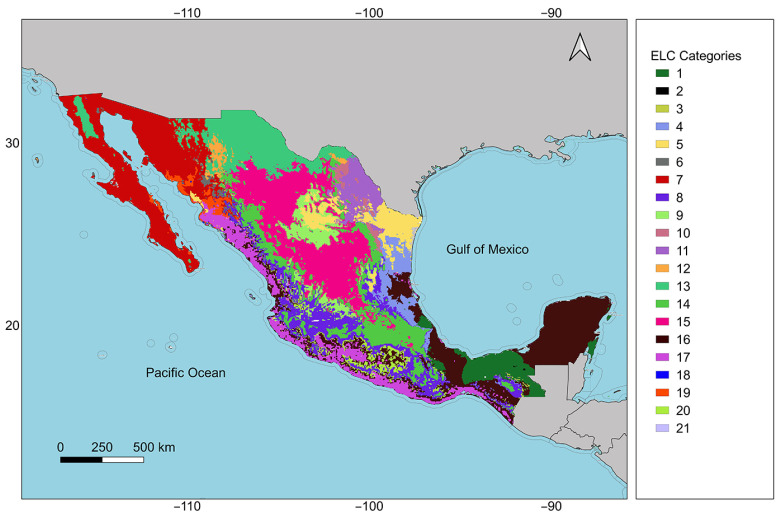
Ecogeographic land characterization (ELC) map of 21 categories reflecting adaptive scenarios for *Melothria pendula* in Mexico.

**Figure 3 plants-13-02572-f003:**
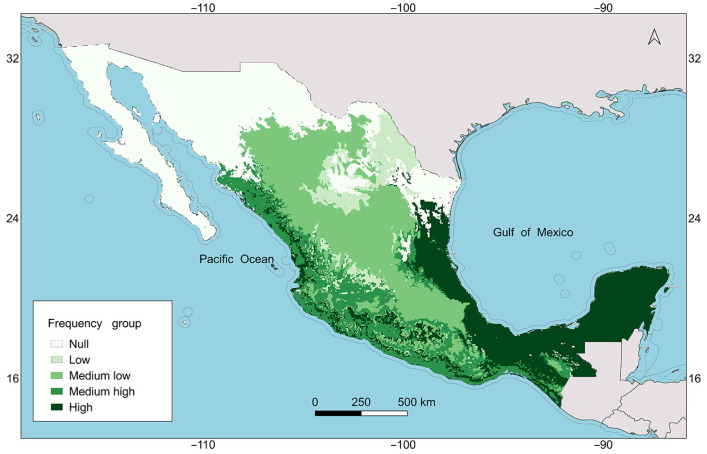
Location of category groups on the ecogeographic land characterization (ELC) map based on the frequency of occurrences of *Melothria pendula*.

**Figure 4 plants-13-02572-f004:**
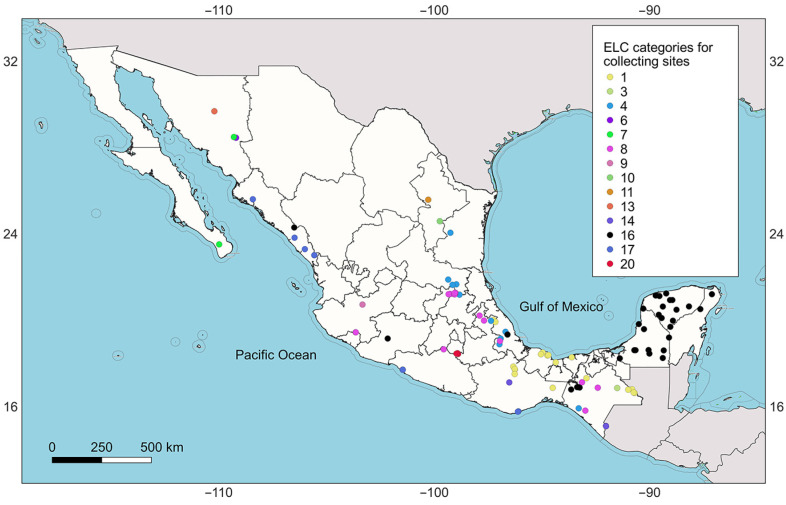
Location of the 111 proposed collecting sites for the establishment of a genebank with high ecogeographic representativeness of *Melothria pendula*.

**Figure 5 plants-13-02572-f005:**
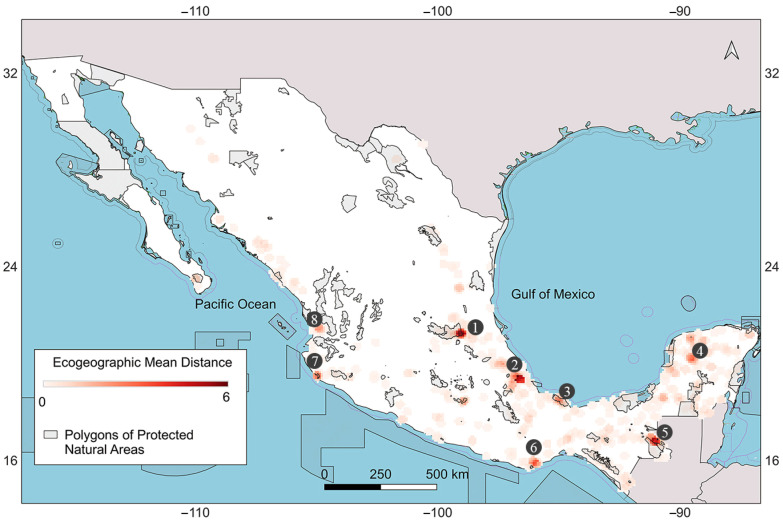
Map of ecogeographic diversity of *Melothria pendula*. Zones 1, 2, and 5 correspond to the areas of highest diversity, whereas zones 3, 4, 6, 7, and 8 reach medium-high values.

**Table 1 plants-13-02572-t001:** Ecogeographic variables selected for their importance in the distribution of *Melothria pendula* in Mexico.

Abiotic Component	Variable Names	CAPFITOGEN3 Variable Code
Climatic	Mean temperature of the coldest quarter	BIO11
	Mean temperature of the driest quarter	BIO9
	Precipitation of the coldest quarter	BIO19
Edaphic	Bulk density top	t_bulk_dens
	Cation exchange capacity top	t_cecsol
	Organic carbon density top	t_oc_dens
	Soil pH KCl top	t_ph_kcl
Geophysical	Elevation	alt
	Solar monthly radiation February	srad_2
	Solar monthly radiation April	srad_4
	Solar monthly radiation July	srad_7

**Table 2 plants-13-02572-t002:** Categories of the ecogeographic land characterization (ELC) map grouped by quartiles of frequency of occurrence of *Melothria pendula* populations and calculation of initial and final quotas (populations to be collected per category) with their respective percentages.

Ecogeographic Category of ELC Map of Mexico	Quartile Group by Frequency	Number of Occurrences and Percentage of Total Distribution	Initial Quotas by P Strategy and 10% Sampling Intensity	Final Quotas and Percentage of Total Future Collections
1	High	183 (17.2%)	18.3	18 (16.2%)
2	Null	0 (0%)	0	0 (0%)
3	Low	1 (0.1%)	0.1	1 (0.9%)
4	High	108 (10.2%)	10.8	11 (9.9%)
5	Null	0 (0%)	0	0 (0%)
6	Low	1 (0.1%)	0.1	1 (0.9%)
7	Medium-low	15 (1.4%)	1.5	2 (1.8%)
8	High	154 (14.5%)	15.4	15 (13.5%)
9	Low	1 (0.1%)	0.1	1 (0.9%)
10	Medium-low	8 (0.8%)	0.8	1 (0.9%)
11	Medium-low	3 (0.3%)	0.3	1 (0.9%)
12	Null	0 (0%)	0	0 (0%)
13	Low	1 (0.1%)	0.1	1 (0.9%)
14	Medium-high	28 (2.6%)	2.7	3 (2.7%)
15	Null	0 (0%)	0	0 (0%)
16	High	418 (39.3%)	41.8	42 (37.8%)
17	Medium-high	108 (10.2%)	10.8	11 (9.9%)
18	Null	0 (0%)	0	0 (0%)
19	Null	0 (0%)	0	0 (0%)
20	Medium-high	34 (3.2%)	3.4	3 (2.7%)
21	Null	0 (0%)	0	0 (0%)

## Data Availability

Data are available upon request.

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
