# Peer review of "Spatial Analysis of the Ecogeographic Diversity of Wild Creeping Cucumber (Melothria pendula L.) for In Situ and Ex Situ Conservation in Mexico"

_plants, 2024, doi:10.3390/plants13182572_

Round 1
Reviewer 1 Report
Comments and Suggestions for Authors
This is a fine paper, worthy of publication. It is a bit wordy, so I’ve tried to polish up your English and condense it a bit. All my proposed edits are highlighted in yellow in the attached file.
I have only a few points that need special attention. Most are quite minor (only #3 is major).
1. On Lines 109, 118, and 279, please replace the letter “x” with the multiplication sign “×”.
2. On Lines 153-154, please include at least one literature citation to support your strategy.
3. In Figure 4, you map 111 proposed collection sites. However, you never present your strategy for dealing with the fact that 10%, 20%, or even 50% of your proposed sites may be failures. Urbanization and other land-use changes, climate change, and just pure luck can undermine plant collection efforts based on historical records. On Lines 346-350, you mention possible adjustments to your collection strategy. But these are NOT Materials and Methods. That passage (Lines 346-350) needs to be moved to a new paragraph in your Discussion (perhaps inserted after Line 252), where you describe in detail the strategies you would follow in the event that a significant proportion of your targeted sites do not result in viable collections for a new genebank.
4. On Line 188, I inserted reference 46 to support your statement. This will require the renumbering of all references (as may recommendations #2 and #6).
5. On Line 251, your phrase “some of them” is unclear to me. Please clarify.
6. On Lines 330-333, please include at least one or two literature citations to support these statements (absence and priority for representation).
7. Can you please include the authors at the beginnings of Reference 25 (Line 469) and of Reference 42 (Line 506)?

Comments on the Quality of English LanguageNoted in Comments to Authors (above). I have attempted to polish and tighten up your manuscript in the attached file.
Author Response
Summary
Thank you very much for taking the time to review this manuscript and improve the English writing. Please find the detailed responses below and the corresponding corrections in track changes in the re-submitted file.
Comment 1: This is a fine paper, worthy of publication. It is a bit wordy, so I’ve tried to polish up your English and condense it a bit. All my proposed edits are highlighted in yellow in the attached file.
Response 1: We are really thankful for your assistance in polishing up the English writing. All sentences were corrected/improved using the track changes tool and following the Reviewer’s suggestions on the PDF.
Comment 2: On Lines 109, 118, and 279, please replace the letter “x” with the multiplication sign “×”.
Response 2: Thank you for pointing this out. We have replaced the letter “x” with the multiplication sign “×” in the lines you highlighted. These changes can be found on page 3, line 117; page 4, line 140, and page 10, line 331.
Comment 3: On Lines 153-154, please include at least one literature citation to support your strategy.
Response 3: Agree. However, we did not introduce a citation to explain the context of the use of the P allocation strategy in this section; instead, we did so in the Materials and Methods section, where a new citation illustrating the P allocation strategy can be found. Please check page 11, line 393.
Comment 4: In Figure 4, you map 111 proposed collection sites. However, you never present your strategy for dealing with the fact that 10%, 20%, or even 50% of your proposed sites may be failures. Urbanization and other land-use changes, climate change, and just pure luck can undermine plant collection efforts based on historical records. On Lines 346-350, you mention possible adjustments to your collection strategy. But these are NOT Materials and Methods. That passage (Lines 346-350) needs to be moved to a new paragraph in your Discussion (perhaps inserted after Line 252), where you describe in detail the strategies you would follow in the event that a significant proportion of your targeted sites do not result in viable collections for a new genebank.
Response 4: Thanks for pointing this out. However, these lines are simply an example of how the combination of the allocation strategy and sampling intensity works. It is in the Materials and Methods section because it illustrates the methodology with an example, but if we move it, the example will not contribute to the context of the results. The authors would prefer to keep the paragraph in the original part of the manuscript.
Comment 5: On Line 188, I inserted reference 46 to support your statement. This will require the renumbering of all references (as may recommendations #2 and #6).
Response 5: Thanks for adding the reference to support the statement. All references were renumbered as suggested by the reviewer.
Comment 6: On Line 251, your phrase “some of them” is unclear to me. Please clarify.
Response 6: The phrase has been replaced by “some of the adaptive scenarios” to clarify the meaning of the sentence. This change can be found on page 9, line 299.
Comment 7: On Lines 330-333, please include at least one or two literature citations to support these statements (absence and priority for representation).
Response 7: Agree. We have included a citation to support the statements (Parra-Quijano, M.; Iriondo, J.M.; Torres, E. Improving representativeness of genebank collections through species distribution models, gap analysis and ecogeographical maps. Biodivers. Conserv. 2012, 21, 79–96, https://doi.org/10.1007/s10531-011-0167-0). This change can be found on page 11, line 391.
Comment 8: Can you please include the authors at the beginnings of Reference 25 (Line 469) and of Reference 42 (Line 506)?
Response 8: Thanks for pointing this out. However, we are unable to include the authors at the beginnings of these references since we are following the journal’s guidelines to cite information from websites, which shall not include the authors’ names.
Comment 9: Some words in the references need to be italicized.
Response 9: Agree. We have italicized all the words highlighted by the reviewer. They are highlighted in the References section.
Reviewer 2 Report
Comments and Suggestions for Authors
The manuscript “Spatial analysis of the ecogeographic diversity of wild creeping cucumber (Melothria pendula L.) for in situ and ex situ conservation in Mexico” which investigates the relationships between the distribution of species Melothria pendula L. in relation to key environmental factors, with a view to promoting its conservation in situ and ex situ, is well structured and well written.
Figures and tables are adequate and fairly well done. In my opinion, there are no major revisions to be made and it can be published after solving some minor problems:
- in figure 1 (a) the blue symbols are poorly identifiable, I would change the blue color to a brighter one;
- in figure 2 the colors are many, some numbers could be given in the figure to help identify ELCs;
- in figure 5 the numbers are hard to read, maybe you could try increasing the size of the numbers.
Author Response
Comment 1: The manuscript “Spatial analysis of the ecogeographic diversity of wild creeping cucumber (Melothria pendula L.) for in situ and ex situ conservation in Mexico” which investigates the relationships between the distribution of species Melothria pendula L. in relation to key environmental factors, with a view to promoting its conservation in situ and ex situ, is well structured and well written.
Response 1: The authors would like to thank Reviewer 2 for taking the time to review this manuscript. Please find the corresponding changes in the figures in the re-submitted file.
Comment 2: Figures and tables are adequate and fairly well done. In my opinion, there are no major revisions to be made and it can be published after solving some minor problems:
In figure 1 (a) the blue symbols are poorly identifiable, I would change the blue color to a brighter one.
Response 2: Agree. The changes requested by the reviewer were made and we changed the color blue for green. The new figure can be found on page 3, line 121 (Figure 1).
Comment 3: In figure 2 the colors are many, some numbers could be given in the figure to help identify ELCs.
Response 3: Thanks for your suggestion. Although there are many colors in this figure, we have published ELC maps with far more than 21 categories (different colors) before. The reason is that these figures are merely representations; this image cannot be taken as a functional element. For that purpose, the GIS format map can be requested from the authors, but the map image is only illustrative.
Comment 4: In figure 5 the numbers are hard to read, maybe you could try increasing the size of the numbers.
Response 4: Agree. The changes requested by the reviewer were made. The new figure can be found on page 8, lines 232 and 233 (Figure 5).
Round 2
Reviewer 1 Report
Comments and Suggestions for Authors
The new version is much improved. However, you did not address the single, most important point that I raised in the initial review (to quote): "In Figure 4, you map 111 proposed collection sites. However, you never present your strategy for dealing with the fact that 10%, 20%, or even 50% of your proposed sites may be failures. Urbanization and other land-use changes, climate change, and just pure luck can undermine plant collection efforts based on historical records. On Lines 346-350, you mention possible adjustments to your collection strategy. But these are NOT Materials and Methods. That passage (Lines 346-350) needs to be moved to a new paragraph in your Discussion (perhaps inserted after Line 252), where you describe in detail the strategies you would follow in the event that a significant proportion of your targeted sites do not result in viable collections for a new genebank."
I understand your desire to keep "old" Lines 346-350 in the Materials and Methods. That's fine, as long as you add a detailed discussion of strategies that you would follow if a significant proportion of your targeted sites don't results in viable collections into your Discussion as a new paragraph after Line 370. This is practical information that readers clearly need to know about.
Author Response
Comment 1: The new version is much improved. However, you did not address the single, most important point that I raised in the initial review (to quote):"In Figure 4, you map 111 proposed collection sites. However, you never present your strategy for dealing with the fact that 10%, 20%, or even 50% of your proposed sites may be failures. Urbanization and other land-use changes, climate change, and just pure luck can undermine plant collection efforts based on historical records.
Response 1:
Thank you for pointing this out. This is a risk inherent to any germplasm collection design and can only be properly assessed at the time of collection. The expedition planning methodology presented here does not seek to eliminate this risk but rather to ensure that collections (if they occur) at certain sites provide maximum ecogeographic representativeness for a future gene bank, or for enhancing an existing one. As a general rule, previous studies on germplasm collection do not directly address this risk. Instead, they rely on final-stage masking maps that typically incorporate data on water bodies, urban centers, or land use, assuming that the target plants either cannot or must be present in these areas. The following study provides an example:
Jarvis, A., Williams, K., Williams, D., Guarino, L., Caballero, P. J., & Mottram, G. (2005). Use of GIS for optimizing a collecting mission for a rare wild pepper (Capsicum flexuosum Sendtn.) in Paraguay. Genetic Resources and Crop Evolution, 52, 671-682.
This study uses a Land Cover Classification mask map to identify forested areas where the target plant is likely to occur, with the aim of narrowing down areas for exploration. It seeks to determine collection sites through Species Distribution Models (SDMs), which often indicate broad regions of high species occurrence probability. Reducing this uncertainty is key, making the use of mask maps in the final stages a necessary step.
In our study, the collecting design is not based on models or areas with a high probability of finding germplasm, but rather on actual occurrences observed at specific points. This approach helps reduce the risk of not finding the desired populations, especially when the occurrence data are relatively recent. Therefore, the first step in our design is to evaluate the georeferencing quality of the gathered occurrence sites. This evaluation not only checks for accuracy and consistency in the coordinates and locality descriptions but also assigns higher quality value to occurrences that, according to global land use maps (1 km resolution), fall within areas compatible with the presence of the target plant species, whether cultivated or wild. By filtering these occurrences, we ensure that only those meeting predetermined quality thresholds are considered in subsequent steps and selected as potential collection sites. While this method provides a higher likelihood of locating the desired populations, we acknowledge that neither this nor any other methodology to date can guarantee a 100% success rate in finding the target population.
Comment 2:
On Lines 346-350, you mention possible adjustments to your collection strategy. But these are NOT Materials and Methods. That passage (Lines 346-350) needs to be moved to a new paragraph in your Discussion (perhaps inserted after Line 252), where you describe in detail the strategies you would follow in the event that a significant proportion of your targeted sites do not result in viable collections for a new genebank.
I understand your desire to keep "old" Lines 346-350 in the Materials and Methods. That's fine, as long as you add a detailed discussion of strategies that you would follow if a significant proportion of your targeted sites don't results in viable collections into your Discussion as a new paragraph after Line 370. This is practical information that readers clearly need to know about.
Response 2:
We agree with the reviewer’s suggestion. As a result, we have added a paragraph in line 411, where we believe it adds more clarity and relevance, as follows:
To reduce the risk of not locating target populations in the field due to factors such as urbanization, climate change, or land use changes, the first step in our design was to assess the georeferencing quality of the gathered occurrence sites. We assigned a quality score to occurrences that, according to global land use maps like GlobCover (1 km resolution), are located in areas compatible with the presence of the intended plant type (cultivated or wild) for collection (see the GEOQUAL tool, [23]). By filtering occurrences, only those that meet predetermined quality criteria can proceed to subsequent steps and be considered as potential collection sites. While this design increases the likelihood of successfully locating the desired populations, field validation remains, to date, the only definitive way to confirm the occurrence of the target population.

Round 3
Reviewer 1 Report
Comments and Suggestions for Authors
Dear authors, I see that you do not wish to get into a discussion of failures in the field and appropriate contingency analyses. Be that as it may, I wish you would have. It would make for a practical addition to this paper. However, I think the paper is now at a state where it is ready for publication.